# Chimeric Antigen Receptor T Cell Therapy for Pancreatic Cancer: A Review of Current Evidence

**DOI:** 10.3390/cells13010101

**Published:** 2024-01-04

**Authors:** Agata Czaplicka, Mieszko Lachota, Leszek Pączek, Radosław Zagożdżon, Beata Kaleta

**Affiliations:** 1Department of Internal Medicine and Gastroenterology, Mazovian “Bródnowski” Hospital, 03-242 Warsaw, Poland; agata.sieczkowska23@gmail.com; 2Laboratory of Cellular and Genetic Therapies, Medical University of Warsaw, 02-091 Warsaw, Poland; mieszko.lachota@wum.edu.pl (M.L.); radoslaw.zagozdzon@wum.edu.pl (R.Z.); 3Department of Clinical Immunology, Medical University of Warsaw, 02-006 Warsaw, Poland; leszek.paczek@wum.edu.pl

**Keywords:** adoptive cell therapy, CAR T cells, chimeric antigen receptor, immunotherapy, pancreatic cancer

## Abstract

Chimeric antigen receptor (CAR) T-cell therapy has revolutionized the treatment of malignant and non-malignant disorders. CARs are synthetic transmembrane receptors expressed on genetically modified immune effector cells, including T cells, natural killer (NK) cells, or macrophages, which are able to recognize specific surface antigens on target cells and eliminate them. CAR-modified immune cells mediate cytotoxic antitumor effects via numerous mechanisms, including the perforin and granzyme pathway, Fas and Fas Ligand (FasL) pathway, and cytokine secretion. High hopes are associated with the prospective use of the CAR-T strategy against solid cancers, especially the ones resistant to standard oncological therapies, such as pancreatic cancer (PC). Herein, we summarize the current pre-clinical and clinical studies evaluating potential tumor-associated antigens (TAA), CAR-T cell toxicities, and their efficacy in PC.

## 1. Introduction

Pancreatic cancer (PC) is the 12th most common cancer and the 7th leading cause of cancer-related death worldwide [1]. This type of malignancy has a poor prognosis due to delayed diagnosis and limited therapeutic options. The 5-year relative survival rate for PC is 12% [2]. Therefore, the early detection and identification of molecular therapeutic targets improve the prospects for long-term survival. According to data published in 2023 by the National Cancer Institute, 90% of pancreatic cancers are pancreatic ductal adenocarcinomas (PDACs), which are predicted to be the second leading cause of cancer-related death by 2030 [3,4].

PC is associated with numerous risk factors, including age, sex, tobacco smoking, overweight and obesity, hyperglycemia, insulin resistance, type 2 diabetes, pancreatitis, as well as genetic factors (mutations in *BRCA1*, *BRCA2*, *PALB2*, *ATM*, *STK11/LKB1*, *P16INK4A/CDKN2A*, *KRAS5*, and *TP53* genes) [5]. Gene expression analyses allowed the identification of four molecular subtypes of PDAC with different biological traits and potential subtype-specific therapeutic options: (1) squamous; (2) pancreatic progenitor; (3) immunogenic; and (4) aberrantly differentiated endocrine exocrine (ADEX) [6].

PC treatment depends on the stage of the disease, molecular subtype, and location of the cancer. Nowadays, for resectable and borderline resectable tumors, surgery followed by adjuvant chemotherapy (with gemcitabine plus capecitabine) with or without radiotherapy represents the gold standard treatment. In case of unresectable tumors, neoadjuvant chemotherapy with FOLFIRINOX (oxaliplatin, irinotecan, leucovorin, and 5-fluorouracil) is utilized. For metastatic PC, palliative chemotherapy is the only treatment option [7,8]. Progress toward effective first-line therapy has been slow. According to data from clinical trials, up to 86% of patients need a second-line chemotherapy with gemcitabine plus nanoparticle albumin-bound paclitaxel or nanoliposomal irinotecan plus 5-fluorouracil [9].

## 2. Immunotherapy Challenges in Pancreatic Cancer

Cancer immunotherapy, especially with immune checkpoint inhibitors (ICIs), has revolutionized the field of oncology [10]. However, in the case of PC, this therapy is limited due to specific features of the tumor microenvironment (TME), including dense desmoplasia and immunosuppression [11,12,13].

PC creates an immunosuppressive TME favorable for its growth. Cancer cells have low expression of major histocompatibility complex (MHC) class I molecules, which leads to decreased activation of cytotoxic CD8^+^ T cells, lower secretion of perforin and granzyme, as well as decreased expression of IC molecules. Moreover, in PC, immunosuppressive regulatory T cells (Treg) with cytotoxic T-lymphocyte-associated antigen 4 (CTLA-4) on their surface are abundant [14]. Other immunosuppressive cells present in TME are tumor-associated macrophages (TAMs), which inhibit the infiltration of CD8^+^ T cells into the tumor [15]. In addition, M2-like macrophages secrete cytokines and growth factors, which stimulate cancer cell proliferation and metastases [16].

The differentiation of pancreatic stellate cells and epithelial, endothelial, as well as mesenchymal stem cells into myofibroblast-like cells results in the formation of stromal desmoplasia [15], which is a mechanical barrier limiting the infiltration of immune cells [17]. Desmoplasia comprises up to 90% of the tumor volume and is associated with worse outcomes of treatment in patients with PC [18].

ICIs are drugs that target immunologic receptors, which are negative regulators of T-cell immune function. Various ICIs have been approved for the treatment of numerous cancers, including anti-programmed death receptor 1 (PD-1) antibodies (Nivolumab, Pembrolizumab, and Cemiplimab), anti-programmed cell death ligand (PDL-1) antibodies (Atezolimumab, Durvalumab, and Avelumab), and anti-cytotoxic T-lymphocyte-associated protein 4 (CTLA-4) antibodies (Ipilimumab and Tremelimumab) [19]. PD-1/PDL-1 and CTLA-4 blockade restores proinflammatory TME, enhances T-cell proliferation and survival, and increases the production of cytokines, including interleukin (IL)-2, interferon (IFN)-γ, and tumor necrosis factor (TNF)-α [20,21].

As mentioned above, immunotherapy is effective in treating a variety of cancers; however, it is not associated with improved overall survival in PC patients. This could be explained by the specific TME, low tumor mutational burden, and, consequently, the low level of cancer neoantigens [22]. Another explanation may be the existence of different PD-1/PD-L1 or CTLA-4 immune checkpoints in PC. Indeed, alternative immune checkpoints, e.g., from the carcinoembryonic antigen (CEACAM) family of surface molecules [23,24], have been suggested for dominant immunosuppressive roles in pancreatic cancer. Therefore, novel therapeutic strategies are urgently needed to improve the outcomes of patients with PC. A promising alternative could be immunotherapy with chimeric antigen receptor-engineered T cells (CAR-T). However, it has been emphasized that CAR-T cell therapy success in PC treatment is hindered by TME [25]. Stromal desmoplasia is a barrier that prevents the infiltration of cytotoxic CAR-T cells into the tumor. In addition, the dysregulated tumor vasculature contributes to hypoxia, which in turn decreases CAR-T cell extravasation into the TME [14,15,16,25]. Another factor that limits CAR-T cell therapy efficacy is the immunosuppressive profile of cytokines and chemokines in TME, which, as mentioned above, recruits various immunosuppressive cells, including Treg lymphocytes, TAMs, and M2-like macrophages. The effector function of CAR-T cells can also be inhibited by metabolites present in TME, including lactate and kynurenine [26,27,28,29]. Cancer cells compete with CAR-T cells for glucose, contributing to reduced CAR-T effectiveness and exhaustion. Exhausted CAR-T cells are characterized by impaired glycolysis, decreased proliferation, and an increased expression of inhibitory receptors, which leads to weakened antitumor activity [25,30].

To overcome the limitations of CAR-T cell therapy in PC, several strategies are being developed and tested in pre-clinical and clinical studies (summarized in the last part of this manuscript) [25,26,27,28,29,30].

## 3. CAR-T Cells

### 3.1. Structure and Evolution of CAR-T Cells

CAR-T cell-based therapy has revolutionized the treatment of malignant and non-malignant disorders [31,32]. CARs are synthetic transmembrane receptors expressed on genetically modified T lymphocytes, which are able to recognize specific surface antigens on target cells and eliminate them [33]. Currently, six CAR-T cell therapies are available commercially following approvals from Food and Drug Administration (FDA) and European Medicines Agency (EMA): tisagenlecleucel, axicabtagene ciloleucel, brexucabtagene autoleucel, lisocabtagene maraleucel, idecabtagene vicleucel, and relmacabtagene [34]. A number of others are available at the national level. The CAR-T cell manufacturing process is associated with the collection of a patient’s T lymphocytes, their in vitro activation, genetic modification, and expansion. Next, CAR-T cells are infused back into the patient’s bloodstream.

CARs are composed of four functional regions: (1) the extracellular domain, which recognizes and binds target antigens via a single-chain variable fragment (scFv) or variable domain on a heavy chain (VHH) antibodies; (2) the hinge, which provides the flexibility of the receptor; (3) the transmembrane domain composed of CD28, CD8α, CD4, or CD3ζ proteins, which links the extracellular domain with the intracellular signaling domain and is responsible for stabilization; and (4) the endodomain, which is the functional part of the CAR. Usually, activation is mediated by the ζ-chain of CD3 cluster (CD3ζ)-derived immunoreceptor tyrosine-based activation motifs (ITAMs) [35,36].

Currently, CARs are classified into five generations [37]. The first generation contains only the antigen recognition domain and CD3ζ activation domain [38]. The second-generation CARs are equipped with one co-stimulatory molecule (CD28 or 4-1BB), which stimulates their proliferation and survival [39]. In the third generation of CARs, apart from either of two co-stimulatory domains (CD28 or 4-1BB), additional domains can be included (CD40 and OX40) [40]. This generation has a stronger ability to induce T cell proliferation. The fourth CARs generation, also called TRUCK (T cells redirected for antigen-unrestricted cytokine-initiated killing), contains genes encoding immunomodulatory cytokines, such as IL-12 and IL-15, thus being able to recruit other immune cells [41]. The fifth generation, or next generation, of CARs can be equipped with a range of co-stimulatory domains, signal transducers, and transcription activators or logic-gate switches, which are intended to increase the safety and efficiency of immunotherapy [42].

### 3.2. Antitumor Mechanisms of CAR-T Cells

To control cancer cells, CAR-T cells bind to the specific antigen on the surface of the target cell and form a non-classical immune synapse (IS) [43]. IS is an organized structure composed of supramolecular activating clusters (SMACs). The central SMAC consists of the T cell receptor (TCR) and the Lck kinase and provides signaling, as well as termination of activating signals [44]. The central SMAC is surrounded by the peripheral SMAC composed of lymphocyte function associated antigen-1 (LFA-1) and provides adhesion. External distal SMAC is composed of CD43 and CD45 proteins [43,44,45,46]. It was demonstrated that a few minutes after IS formation, CAR-T cells mediate their cytotoxic antitumor effects via various mechanisms, including the perforin and granzyme pathway, Fas and Fas Ligand (FasL) pathway, and cytokine secretion [44,45,46] (Figure 1).

The perforin–granzyme pathway is used by cytotoxic T cells (CTCs) to eliminate transformed cells [47]. CTCs contain granules with perforin and granzymes. Upon IS formation, granules are moved to the interface, where they dock and fuse with the plasma membrane and release cytolytic perforin. On the target cell, perforin oligomerizes and forms pores which allow proapoptic granzyme entry. Next, granzymes activate a caspase-dependent and independent cell death [48,49].

Another pathway associated with the cytotoxic effects of CAR-T cells is the Fas/FasL axis [50]. Fas (CD95) is a type I membrane protein containing the death domain (DD) necessary for apoptosis. This cell-death process is induced by the interaction of Fas with its ligand—FasL [51]. FasL (CD95L) is a type II membrane protein. Its interactions with Fas lead to the recruitment of the adaptor protein Fas-associated death domain (FADD), pro-caspase 8 binding, formation of the death-inducing signaling complex (DISC), and activation of effector caspase-3 [51].

CAR-T cells also mediate their anticancer effects via the secretion of cytokines, which are able to increase CAR-T cell activity, restore stromal sensitization (for example, by increasing IFN-γ receptor expression), and modify TME, including polarization of macrophages toward M1 phenotype [50,51,52,53].

## 4. CAR-T Cell Therapy in Pancreatic Cancer

CAR-T cell-based therapy is nowadays a promising therapeutic option for numerous hematological malignancies, including acute lymphoblastic leukemia, chronic lymphocytic leukemia, Richter’s syndrome, lymphoma, multiple myeloma, and acute myeloid leukemia [54,55]. However, it is not approved by the FDA for the treatment of solid tumors. Nevertheless, high hopes are associated with the prospective use of CAR-T strategy against solid cancers, especially the ones resistant to standard oncological therapies, such as PC. Indeed, current pre-clinical and clinical studies evaluate potential tumor-associated antigens (TAA), cancer markers, CAR-T cell toxicities, and efficacy in PC [56,57].

### 4.1. Studies Performed in Cell Culture and Animal Models

Numerous studies on the use of CAR-T cells in PC therapy were conducted in animal or human cell lines models and aimed at identifying suitable TAA, as well as evaluating CAR-T cells toxicity and efficacy.

One of the cell lines used as a model for studying the biology of PC is AsPC-1—human pancreatic adenocarcinoma cell line—which expresses such TAA as carcino-embryonic antigens (CEA, also referred to as CEACAM5) and mesothelin (MSLN). This line was used by Zhang et al. [58] to assess the effectiveness of the dual-receptor CAR-T cells (dCAR-T) specific for CEA and MSLN. This construct contains two separate domains: CEA-CD3ζ and MSLN-4/1BB. It was demonstrated that dCAR-T exerted high cytotoxic activity against target cells. Moreover, it was revealed that dCAR-T cells significantly inhibited AsPC-1 cell growth, provided proliferation, and secreted high levels of IL-2, IL-6, TNF-α, and IFN-γ in a mouse model of PC.

In another study, the antitumor activity of hYP218 CAR-T cells was evaluated in vitro using mesothelioma cell lines (NCI-Meso29 and NCI-Meso63) and in vivo in NSG mice with mesothelin-expressing PC (KLM-1) [59]. hYP218 CAR-T cells, which target a membrane-proximal epitope, showed cytolytic activity toward NCI-Meso29 and NCI-Meso63 cancer cells and secreted high levels of IL-2, TNF-α, and IFN-γ. In addition, it was shown that a single infusion of hYP218 CAR-T cells into mice resulted in significant tumor regression.

In 2017, Chmielewski and Abken engineered IL-18-secreting CAR-T cells to induce proinflammatory response in CEA-positive pancreatic tumors in immunocompetent mice [60]. It was found that IL-18 CAR-T cells expressed granzyme and perforin and induced Th1 polarization in the TME. CAR-T cell therapy reduced the number of M2 macrophages (responsible for cancer metastasis), increased the number of NKG2D^+^ cells, and decreased the number of immunosuppressive CD4^+^FoxP3^+^ Treg cells. To verify if IL-18 CAR-T cell anticancer therapy is effective for an extended period of time, 20 days after primary CAR-T cells and tumor cells injection, mice were inoculated for the second time. It was revealed that the therapy prevented tumor growth, and CAR-T cells were present in the blood of mice for 60 days after construct transfer.

CD70 is a type II transmembrane protein belonging to the TNF superfamily, strongly associated with poor survival in patients with solid tumors. Its expression is associated with TME immunosuppressive footprint and cancer cell proliferation [61]. Jin et al. evaluated CD70 CAR-T cells engineered to express IL-8 (CXCL8) receptors (CXCR1 or CXCR2) to improve their tumor infiltration capability in PC treatment [62]. This idea was supported by other studies that have shown that CXCR2-modified cells improved tumor infiltration capability [63]. In PC, one of the most prominently expressed chemokines is CXCL8, which also plays a significant role in tumor invasion, angiogenesis, and metastasis; therefore, Jin et al. engineered and evaluated CXCR1-CD70 and CXCR2-CD70 CAR-T cells in human pancreatic carcinoma cell line PANC-1, as well as in vivo in mice inoculated with PANC-1.i720 tumor cells. It was demonstrated that CXCR1 and CXCR2-modified CD70 CARs downregulated the expression of exhaustion markers on T cells and upregulated the migration of T cells in the tumor. In mice, both modified CAR-T cells migrated more efficiently to the tumor site, produced granzymes, and reduced tumor size. Moreover, long-lasting immunologic memory was observed [62].

While equipping CAR-T cells with chemokine receptors represents a potential strategy for improving CAR-T efficacy, we recently showed that the ligation of two different chemokine receptors synergistically potentiates cellular migratory response [64]. Our finding indicates that equipping cells with at least two chemokine receptors matching tumor chemokine secretion profile represents a promising strategy for improving tumor homing, additionally decreasing the risk of tumor immune evasion by selective loss of chemokine expression.

Trophoblast cell-surface antigen 2 (Trop-2) is another marker overexpressed in multiple cancers, including PC, and thus a promising target for immunotherapy [65,66]. Therefore, the aim of a study conducted by Zhu et al. [67] was to determine the anticancer activity of Trop2 CAR-T cells. The effect of Trop2 CAR-T cells on cytotoxicity, degranulation, as well as IL-2, IL-4, IL-6, IL-10, IFN-γ, TNF-α, and IL-17A secretion by PC cell lines (ASPC-1, CFPAC-1, and BxPC-3) was analyzed. Next, the group evaluated the toxicity and antitumor effect of these CAR-T cells in a BxPC-3 pancreatic xenograft model. It was demonstrated that Trop2 CAR-T cells lyse target cells in vitro. The strongest effect was observed against BxPC-3 cells. In addition, CAR-T cells upregulated IL-17A, IL-2, TNF-α, and IFN- γ production by BxPC-3 cells. Studies in NSG mice engrafted with the BxPC-3 cells showed complete disappearance of tumor 28 days after Trop2 CAR-T cells infusion and increased concentration of IFN-γ in the blood. Interestingly, the authors did not observe any toxic effects of the treatment, including weight loss or organ damage.

CAR-T cells able to recognize PD-1 have also proven to be effective at PC cell elimination. Parriott et al. [68] constructed chimeric PD1-Dap10-CD3zeta (chPD1) CAR-T cells and analyzed their anticancer potential in both in vitro and in vivo studies. The inclusion of Dap10 (a co-stimulatory receptor) in CARs enhanced the response of T effector lymphocytes, enhanced cytokines production, and T cell differentiation into memory precursor cells. It was shown that chPD1 CAR-T cells were cytotoxic against PC cell lines Pan02 and TGP49 and increased the synthesis of pro-inflammatory cytokines, including IL-2, IL-17, IL-21, TNF-α, IFN-γ, and granulocyte–macrophage colony-stimulating factor (GM-CSF). In vivo studies were conducted in C57BL/6 mice injected with Pan02 tumor cells. Two doses of chPD1 CAR-T cells were inoculated 5 and 8 days after tumor injection. It was demonstrated that tumor burden was significantly decreased.

Another potential target in the immunotherapy of solid tumors is NK group 2D (NKG2D) receptor [69]. Therefore, Gao et al. engineered CAR-T cells with NKG2D and shRNA-4.1R to evaluate its effects in PC [70]. 4.1R is a cytoskeletal protein that plays a significant role in immunomodulation and cancer development [71]. It was found that 4.1R deletion in NKG2D CAR-T cells is related to their higher cytotoxicity in vitro against PC cells (SW1990, CAPAN2, and PANC28 cells) in a dose-dependent manner. Moreover, in vivo analyses conducted in NSG mice inoculated with PANC28/luc demonstrated that CAR-T cells injected 10 days after cancer cell implantation caused tumor regression.

One of the reasons for the limited long-term response to CAR T-cell therapy in solid tumors is the poor persistence of these cells in vivo [72]. It was found that both the expansion and persistence of CAR-T cells are associated with the type of intracellular domain [73]. Therefore, Guedan et al. [74] incorporated the inducible costimulator (ICOS) intracellular domain into CAR and compared its function and persistence with CARs containing the CD3ζ, CD3ζ/CD28, and the 4-1BB intracellular domains. It was demonstrated that ICOS-based CAR-T cells increased IL-17A, IL-17F, IFN-γ, and IL-22 production. Moreover, in the mouse model, ICOS CARs mediated antitumor responses and showed enhanced persistence compared with CD28- or 4-1BB-based CAR-T cells. In another study, the group demonstrated that the ICOS domain enhanced the persistence of CAR-expressing CD4^+^ T cells that, in turn, increased the persistence of CD8+ T cells expressing either CD28- or 4-1BB–based CARs. Moreover, it was revealed that the combination of ICOS and 4-1BB domains improved the antitumor effects and increased persistence in NSG mice bearing Capan-2 pancreatic tumors [75].

In another study conducted in a murine model of PC, Luu et al. [76] examined the cytotoxic effects of short-chain fatty acids (SCFAs)—modified CAR-T cells. The group demonstrated that in vitro treatment of CAR-T cells with pentanoate and butyrate increases the function of the mTOR signaling pathway (which is associated with tumors) and inhibits class I histone deacetylase activity (overexpressed in cancer). As a result, the upregulated production of CD25, IFN-γ, and TNF-α was observed. Moreover, the group engineered pentanoate-modified CAR-T cells targeting the receptor tyrosine kinase ROR1 and analyzed their in vivo cytotoxic effects in the murine model of PC. ROR1 is a transmembrane receptor considered as a target for PC therapy [77]. ROR1-expressing Panc02 tumor cells were injected into mice. Five days later, pentanoate-treated or untreated ROR1 CAR-T cells were inoculated. It was found that in mice treated with pentanoate-modified CAR-T cells, the tumor volume was significantly smaller than in animals injected with untreated CAR-T cells. In addition, the number of IFN-γ^+^TNF-α^+^ pentanoate-treated CAR-T cells in tumors was higher when compared to controls.

An interesting approach relies on targeting the molecules with suspected roles as negative immune checkpoints in PC, such as CEACAM5 (CEA) [78], CEACAM6 (CD66c) [79], or CEACAM7 [80] or using cross-reactive anti-CEACAM-CAR-T cells [81]. The potential benefits of such an approach can be at least two-fold: (1) direct cytotoxic action against PC cells and (2) decreasing the immunosuppressive capabilities of PC-related microenvironment. Another interesting report in this context showed that the co-expression of IL-4/IL-15-based inverted cytokine receptors in NKG2D-CAR-T cells can overcome IL-4 signaling in an immunosuppressive PC microenvironment [82].

Generally, despite the fact that the current use of CAR-T therapy is more limited in solid tumors than in hematological malignancies, in vitro studies conducted in human cell lines as well as in animal models gave promising results (Table 1).

### 4.2. Studies Performed in Humans

A comprehensive literature search revealed fewer than ten completed clinical trials that explored CAR-T therapy in PC and had published results. All of them were phase I trials, emphasizing safety and dosing evaluations. The trials employed CTCAE 5.0 for grading adverse events and RECIST 1.1 criteria for assessing tumor responses. In some studies, the researchers also evaluated the peripheral blood cytokine profile, tumor biopsies, and other variables to assess the CAR-T cell persistence, their tumor infiltration capability, and other effects of the therapy (Table 2).

NCT02159716, a phase I study, investigated lentiviral-transduced CAR T-cells targeting MSLN in patients with various chemotherapy-refractory cancers, including PDAC [83]. Treatment involved a single infusion of CAR-T cells. The therapy was generally well tolerated except for one instance of grade 4 toxicity (sepsis) without the cyclophosphamide conditioning. Despite observable expansion in blood and the presence of MSLN CAR-T cells in all PC tumor biopsies, clinical activity was limited. Out of five PC patients, three showed no response, and two exhibited stable disease for 2–3 months. Notably, the scFv used in this study contained murine fragments. An assessment of a fully human anti-MSLN CAR is underway (ClinicalTrials.gov: NCT03054298 and NCT03323944).

Prior to the aforementioned study, the same research group conducted another phase I trial in six metastatic PDAC patients resistant to chemotherapy (NCT01897415) [84]. In this study, autologous T cells were transiently modified to express anti-MSLN CAR. The patients received intravenous CAR-T cells three times a week for three weeks. No instances of cytokine release syndrome, neurologic symptoms, or dose-limiting toxicities were noted. Two out of six patients experienced disease stabilization. The progression-free survival (PFS) periods were 3.8 and 5.4 months. Notably, one patient observed a notable 69.2% volume reduction in liver lesions, with no impact on primary PC.

In another phase I trial (NCT01869166), the researchers explored EGFR CAR-T cells in metastatic PC [85]. The inclusion criterion was EGFR expression above 50% on pre-treatment tumor biopsy. Sixteen recruited patients received a nab-paclitaxel and cyclophosphamide conditioning regimen followed by one to three CAR-T cell infusions. Grade ≥ 3 reversible adverse events, including fever, fatigue, nausea, vomiting, mucosal, cutaneous toxicities, pleural effusion, and pulmonary interstitial exudation, were observed. Of the fourteen evaluable patients, four manifested a partial response, and eight had stable disease for 2–4 months each. The median PFS was 3 months following the first EGFR CAR-T cycle, while the median overall survival (OS) was 4.9 months.

Another phase I study utilized HER2 CAR-T cells in advanced biliary tract cancers and PC (NCT01935843) [86]. The inclusion criterion was HER2 expression above 50% on pre-treatment tumor biopsy. A total of 11 participants underwent one to two cycles of HER2 CAR-T cell infusion after conditioning treatment with nab-paclitaxel and cyclophosphamide. Observed adverse effects included grade 3 acute febrile syndrome, one instance of significant transaminase elevation, as well as one severe but reversible upper gastrointestinal hemorrhage. Additionally, two instances of grade 1–2 delayed fever associated with C-reactive protein (CRP) and IL-6 release were recorded. Among the two evaluable PC patients, stable disease lasting 5.3–8.3 months was achieved.

A phase Ib trial investigated the utility of autologous CT041 CAR-T cells against Claudin 18.2 (CLDN18.2), a selective cell lineage marker expressed in certain gastric and PC (NCT04404595) [87]. Eleven recruited patients (five gastric and six pancreatic) underwent a conditioning regimen before receiving CT041 at doses between 2.5 and 4 × 10^8^ cells. The therapy was generally well tolerated, with no incidence of severe cytokine release syndrome (CRS), immune effector cell-associated neurologic syndrome (ICANS), or significant gastrointestinal adverse events, although all patients experienced CRS at grade 1 or 2. Of the first five PC patients evaluated, two achieved stable disease.

Another phase I trial (NCT03874897) evaluated CT041 CAR-T cells in CLDN18.2 positive tumors [88]. The cells were administered at one of three dosages: 2.5 × 10^8^, 3.75 × 10^8^, or 5.0 × 10^8^ cells. While all patients encountered grade 3 or higher hematologic toxicity, and 94.6% experienced grade 1 or 2 CRS, there were no instances of grade 3 or higher CRS, neurotoxicities, treatment-related deaths, or dose-limiting toxicities. Among the five evaluable PC patients, one exhibited non-responsiveness, three maintained stable disease, and one experienced partial remission.

In a phase I clinical study evaluating CD133 CAR-T cells (NCT02541370), a therapy targeting the cancer stem cell marker CD133, 23 patients, including 7 with stage IV PADCs (grade 2 or 3), were enrolled [89]. The therapy was administered at a cell dose of 0.5–2 × 10^6^/kg. The authors report that repeated infusions prolonged disease stability, particularly in those showing initial tumor reduction. The primary toxicity observed was a self-recovering decrease in hemoglobin/platelet (≤grade 3) occurring 3–5 days post-infusion. Patients also experienced grade 2–4 lymphopenia and notable increases in TNF-α, IL-6, and IL-8 levels post-treatment. Immunohistochemistry of biopsied tissues demonstrated the elimination of CD133^+^ cells post-CAR-T infusions. Out of seven evaluable PC patients, two showed no response, three maintained stable disease for 3–10.25 months, and two exhibited partial remission for 2–4 months.

Initial CAR-T therapy trials in PC illuminated a path of both potential and challenges in treating this malignancy. Safety profiles were generally acceptable, with manageable and reversible adverse events. Although phase I trials focus on safety and dosing evaluations, the efficacy was also being monitored. Partial responses or stable diseases were achieved only in a small fraction of patients, with a large proportion showing no response. Despite these initial advancements, the journey toward establishing effective CAR-T therapies for PC remains to be determined, and success is not guaranteed. Some studies showed poor infiltration of CAR-T cells into the tumor, while others exhibited poor persistence [83,84]. The complex biology of the disease necessitates comprehensive, systematic studies of the molecular mechanisms underlying the failures of currently tested treatments to refine and optimize next-generation CAR-T cells. While there are over 38 active clinical trials of CAR-T therapy in PC, their design is strikingly similar to those that were discussed above [90].

### 4.3. Challenges of PC CAR-T Cell Therapy in Clinical Translation and Potential Strategies to Overcome Limitations

As mentioned above, PC therapy with CAR-T cells is limited due to specific features of the TME. These include (1) stromal desmoplasia, (2) heterogeneous antigen expression, and (3) immunosuppression. Other potential limitations that hamper the efficacy of CAR-T cells in PC are cell-mediated toxicities [11,12,13,14,15,16,17,18,19,20,21]. To improve the clinical efficacy of CAR-T cells in PC, several strategies are being developed [25,26,27,28,29,30] (Figure 2).

Stromal desmoplasia, as well as deregulated vasculature, act as a barrier that prevents adequate delivery of CAR-T cells, impairing their therapeutic potential. One method to overcome these physical limitations is the local (intratumoral) administration of CAR-T cells. However, this method has not been evaluated in PC treatment. Another strategy is the generation of CAR-T cells expressing enzymes against tumor stroma (e.g., heparanase) or chemokine receptors (e.g., CXCR1 and CXCR2) matching tumor chemokines and promoting infiltration [17,62].

One of the biggest limitations of CAR-T-based therapy is antigen escape—downregulation or loss of antigen by tumor cells. Therefore, dual-targeting CAR-T cells (anti-MSLN and anti-CEA) have been engineered and tested in a PC mouse model with satisfactory results [58]. Ko and colleagues analyzed the effectiveness and safety of the infusion of two CAR-T cell constructs targeting CD19 and MSLN in patients with PDAC [91]. The treatment was well tolerated without toxicities. However, a lack of tumor infiltration was observed.

The presence of an immunosuppressive microenvironment is another challenge in the application of CAR-T cells in PC [12,27]. Reversal of the TME is an active but early area of research and includes (1) preconditioning chemotherapy, which deletes immunosuppressive immune cells and increases CAR-T cells persistence [83,92]; (2) combination therapy with ICIs (e.g., anti-PD-1/PDL-1), which protects CAR-T cells from exhaustion and senescence [93]; and (3) generation of CAR-T cells expressing proinflammatory cytokines (e.g., IL-12 and IL-27) [94].

CAR-T cell-associated toxicities are significant challenges of immunotherapy. The most common adverse events are cytokine release syndrome (CRS), immune effector cell-associated neurotoxicity syndrome (ICANS), and on-target, off-tumor toxicity [95]. Therefore various strategies have been developed to reduce the risk of these complications. However, none of them have been studied in PC patients. These include the engineering of CAR-T cells with a decreased affinity of the antigen-binding domain, inhibitory CAR-T cells that recognize specific antigens exclusively on non-malignant cells, and CAR-T cells with “off-switches” [33].

## 5. Conclusions and Future Perspectives

PC is one of the most difficult-to-treat human malignancies. Cancer immunotherapy has revolutionized the field of oncology; however, in the case of PC, this therapy is limited due to specific features of the TME. Research currently focuses on expanding CAR-based therapy treatment of various solid tumors, and data collected in the laboratory studies provided clear evidence that properly designed CAR-T cells can produce a powerful cytotoxic effect against PC cells. Nevertheless, before these results can be effectively translated into clinical benefits for PC patients, CAR-T therapies must be upgraded in three main areas: (1) finding suitable molecular targets to overcome tumor heterogeneity and on-target, off-tumor toxicity issues; (2) ensuring the persistence of CAR-T cells in an active, non-exhausted form within PC; and (3) the targeting and elimination of tumor immunosuppressive cells, including Treg lymphocytes, TAMs, and myeloid-derived suppressor cells (MDSCs), via a combination of CAR-T cells with antibodies/drugs lowering the number of these cells or generation of CAR-T cells directly targeting antigens expressed on these cells.

We believe that the usage of next-generation CAR-T therapies, harboring multiple genetic modifications in order to amplify their effectiveness and combinatory and multimodal treatment strategies, will pave the way for the successful application of CAR-T cells against PC.

## Figures and Tables

**Figure 1 cells-13-00101-f001:**
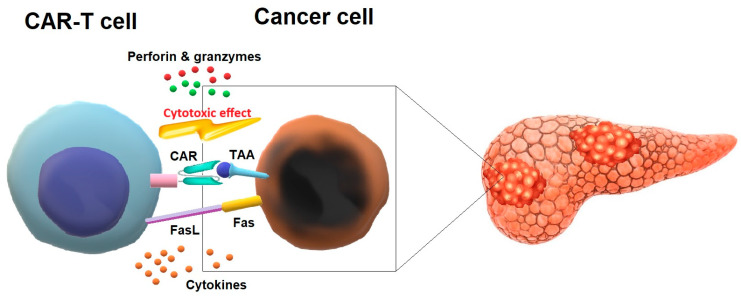
Antitumor mechanisms of chimeric antigen receptor (CAR)-T cells. CAR-T cell recognizes tumor-associated antigens (TAA) on cancer cells and mediates cytotoxic antitumor effects mainly via perforin and granzyme pathway, Fas and Fas Ligand (FasL) pathway, and cytokine secretion.

**Figure 2 cells-13-00101-f002:**
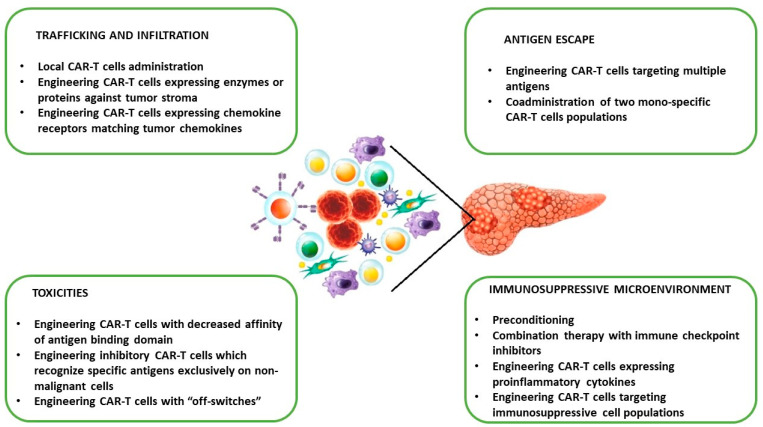
Major limitations of CAR-T cell therapy for pancreatic cancer and potential strategies to overcome limitations.

**Table 1 cells-13-00101-t001:** Pre-clinical studies on CAR-T cell therapy in pancreatic cancer.

Model	Targeted Tumor Antigen	Type of CAR-T Cells	Main Outcomes	Reference
In vitro and in vivo	CEA and MSLN	Dual receptor (anti-CEA and anti-MSLN) CAR-T cells	High cytotoxic activity against target cell line (AsPC-1). Inhibition of tumor growth in a mouse model. High IL-2, IL-6, TNF-α, and IFN-γ secretion in mouse model.	Zhang et al. [58]
In vitro and in vivo	MSLN	Anti-MSLN CAR-T cells	Cytotoxic activity against target cell lines (NCI-Meso29 and NCI-Meso63). High IL-2, TNF-α, and IFN-γ secretion by NCI-Meso63 cell line. Significant tumor regression in a mouse model.	Tomar et al. [59]
In vivo	CEA	IL-18-secreting CAR-T cells	High expression of granzyme and perforin. Increased number of M2 macrophages and NKG2D^+^ Treg cells. Significant tumor regression.	Chmielewski and Abken [61]
In vitro and in vivo	CD70	Anti-CD70 CAR-T cells expressing CXCR1 and CXCR2	Cytotoxic activity against target cell line (PANC-1). Decreased expression of exhaustion markers on T cells and enhanced migration of T cells in the tumor in mice inoculated with PANC-1.i720 tumor cells. High granzyme secretion and reduced tumor size.	Jin et al. [62]
In vitro and in vivo	Trop2	Anti-Trop2 CAR-T cells	Cytotoxic activity against target cell lines (ASPC-1, CFPAC-1, BxPC-3). Upregulated IL-17A, IL-2, TNF-α, and IFN- γ production by BxPC-3 cells. Complete tumor regression and increased IFN-γ in mice inoculated with BxPC-3 tumor cells.	Zhu et al. [67]
In vitro and in vivo	PD-1	PD1-Dap10-CD3zeta CAR-T cells	Cytotoxic activity against target cell lines (Pan02 and TGP49). Increased synthesis of IL-2, IL-17, IL-21, TNF-α, IFN-γ, and GM-CSF. Reduced tumor burden in mice inoculated with Pan02 tumor cells.	Parriott et al. [68]
In vitro and in vivo	NKG2D	NKG2D CAR-T cells with deleted 4.1R protein	4.1R deletion in NKG2D CAR-T cells resulted in higher cytotoxicity against target cell lines (SW1990, CAPAN2, and PANC28). Significant tumor regression in mice inoculated with PANC28 tumor cells.	Gao et al. [70]
In vitro and in vivo	MSLN	CAR-T cells with ICOS	Increased synthesis of IL-17A, IL-17F, IFN-γ, and IL-22 in vitro. Stronger antitumor response in mice inoculated with Capan-2 tumor cells. Enhanced persistence compared with CD28- or 4-1BB-based CAR-T cells.	Guedan et al. [74,75]
In vitro and in vivo	ROR1	SCFAs—modified CAR-T cells	Increased production of CD25, IFN-γ, and TNF-α in vitro. Significant tumor regression in mice inoculated with ROR1+ Pan02 tumor cells.	Luu et al. [76]

CAR, chimeric antigen receptor; CEA, carcino-embryonic antigen; CXCR, chemokine receptor; GM-CSF, granulocyte–macrophage colony-stimulating factor; ICOS, inducible costimulatory; IFN, interferon; IL, interleukin; MSLN, mesothelin; ROR1, receptor tyrosine kinase; SCFAs, short-chain fatty acids; TNF, tumor necrosis factor; Treg cells, regulatory T cells.

**Table 2 cells-13-00101-t002:** Completed clinical trials on CAR-T cell therapy in pancreatic cancer.

Trial Number	Phase	Target	Number of Patients/Treatment	Efficacy	Reference
NCT02159716	I	MSLN	6 PDAC patients with or without cyclophosphamide preconditioning/single CAR-T cell infusion	3/5 evaluable patients with PD; 2/5 patients with SD	Haas et al. [83]
NCT01897415	I	MSLN	6 PDAC patients (information about preconditioning not available)/3 CAR-T cells infusion cycles 3 times/week for 3 weeks	2/3 evaluable patients with SD; 1/3 patients with DP	Beatty et al. [84]
NCT01869166	I	EGFR	16 PDAC patients with nab-paclitaxel and cyclophosphamide preconditioning/single CAR-T cells infusion	4/14 evaluable patients with PR; 8/14 patients with SD; 2/14 patients with DP	Liu et al. [85]
NCT01935843	I	HER2	11 PDAC patients with nab-paclitaxel and cyclophosphamide preconditioning/1–2 CAR-T cells infusion cycles for 3–5 days	2/11 evaluable patients with SD	Feng et al. [86]
NCT04404595	Ib	CLDN18.2	6 PDAC patients with fludarbine, nab-paclitaxel, and cyclophosphamide preconditioning/single CAR-T cells infusion	2/5 evaluable patients with SD	Botta et al. [87]
NCT03874897	I	CLDN18.2	5 PDAC patients: 3 with fludarbine, nab-paclitaxel, and cyclophosphamide preconditioning, 2 with fludarbine, gemcitabine, and cyclophosphamide preconditioning/single CAR-T cells infusion	1/5 evaluable patients with PD; 3/5 patients with SD	Oi et al. [88]
NCT02541370	I	CD133	23 PDAC patients with nab-paclitaxel and cyclophosphamide preconditioning/2–4 CAR-T cells infusion cycles	2/7 evaluable patients with PD; 3/7 patients with SD; 2/7 patients with PR	Wang et al. [89]

CAR, chimeric antigen receptor; CLDN18.2, claudin18.2; DP, disease progression; EGFR, epidermal growth factor receptor; HER2, human epidermal growth factor receptor 2; MSLN, mesothelin; PD, progressive disease; PDAC, pancreatic ductal adenocarcinoma; PR, partial response; SD, stable disease.

## Data Availability

Data sharing is not applicable.

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
