# Peer review of "Chimeric Antigen Receptor T Cell Therapy for Pancreatic Cancer: A Review of Current Evidence"

_cells, 2024, doi:10.3390/cells13010101_

Round 1
Reviewer 1 Report
Comments and Suggestions for Authors
The review is well written and effectively structured, offering a comprehensive depiction of the latest advancements in CAR T cell therapy specifically tailored for pancreatic cancers.
I would expand Chapter 2 to interconnect the potential limitations of CAR T cell therapy with the immunosuppressive Tumor Microenvironment (TME) prevalent in pancreatic cancer. I think this would enhance the review's depth. This linkage could delve into how the unique characteristics of the TME, such as dense stroma, immune cell exclusion, and metabolic reprogramming, may hinder CAR T cell effectiveness.
Minor comments:
- I would suggest incorporating a table summarizing the clinical trials ongoing and adding columns with the differences between the studies (i.e. type of CAR, generation, conditioning regimens...).
Author Response
Dear Reviewer, thank you very much for your constructive comments and recommendations. We have addressed them accordingly in the revised version of the manuscript entitled: „Chimeric antigen receptor T cell therapy for pancreatic cancer: a review of current evidence” as described in details below. We hope that our revisions improved the paper such that Reviewer and Editor now deem it suitable for publication in Cells.
Please find the detailed responses to Reviewer’s comments in the attached file.

Reviewer 2 Report
Comments and Suggestions for Authors
In the manuscript entitled "Chimeric antigen receptor T cell therapy for pancreatic cancer: a review of current evidence" the authors made an effort to provide a thorough analysis of the therapeutic potential of CAR-T cells for pancreatic cancer. The topic is very interesting and the manuscript has been written well. Nonetheless, the manuscript still has to be refined significantly before it can be considered for publication.
Specific concerns:
1. In the Introduction section, authors should discuss immunotherapy, its current status, and CAR-T cell therapy. This would create a continuation to the next section.
2. The statement "Pancreatic cancer (PC) is the 12th most common cancer" in Line 26 lacks a specific reference to support this claim.
3. The cited article (reference number 2) does not provide information on the 5-year relative survival rate for pancreatic cancer is 12%.
4. The mention of specific mutations like "BRCA1, BRCA2, PALB2, ATM, STK11/LKB1, P16INK4A/ CDKN2A, KRAS5, and TP53" in relation to pancreatic cancer is not supported by the cited article (reference number 5).
5. The manuscript could enhance clarity and adherence to standard formatting by consistently providing the full forms of abbreviations, such as PD-1/PDL-1, FDA, EMA, and MDSCs upon their initial mention.
6. The paragraph discussing CAR-T cells effectively presents a concise summary of their structure. In the subsequent section on the mechanism of CAR-T cells, the text mentions key pathways like the perforin-granzyme pathway and the Fas/FasL axis, along with cytokine secretion contributing to their anticancer effects. To enhance comprehension, a visually informative figure should be created to illustrate these mechanisms.
7. In the section of CAR-T therapy in pancreatic cancer provides an insightful overview of various pre-clinical and clinical studies. To enhance the reader's understanding, it would be beneficial to consider the inclusion of a table summarizing key information from the reviewed studies. This table could include details such as the targeted antigens, CAR-T cell constructs, findings in terms of cytotoxic activity and cytokine secretion, as well as outcomes in both in vitro and in vivo models.
8. A comprehensive table summarising all the clinical trials so far should be included.
9. It is recommended to add a section on challenges of CAR-T cell therapy in clinical translation.
10. A critical study of the subject would provide further insights and make the subject more intriguing to readers. Otherwise, the aim of a review is defeated since it devolves into a collection of readily available information.
Comments on the Quality of English Language
Quality of language is good.
Author Response

(The authors gave the same response as above.)
